# Manual Chest Compression versus Automated Chest Compression Device during Day-Time and Night-Time Resuscitation Following Out-of-Hospital Cardiac Arrest: A Retrospective Historical Control Study

**DOI:** 10.3390/jpm13081202

**Published:** 2023-07-28

**Authors:** Wataru Takayama, Akira Endo, Koji Morishita, Yasuhiro Otomo

**Affiliations:** 1Trauma and Acute Critical Care Center, Tokyo Medical and Dental University Hospital of Medicine, 1-5-45, Yushima, Bunkyo-ku, Tokyo 113-0034, Japan; morishita.accm@tmd.ac.jp (K.M.); otomo.accm@tmd.ac.jp (Y.O.); 2Department of Acute Critical Care and Disaster Medicine, Graduate School of Tokyo Medical and Dental University, Tokyo Medical and Dental University, 1-5-45, Yushima, Bunkyo-ku, Tokyo 113-0034, Japan; eraeaccm@tmd.ac.jp; 3Department of Acute Critical Care Medicine, Tsuchiura Kyodo General Hospital, 4-1-1 Otsuno, Tsuchiura 300-0028, Ibaraki, Japan

**Keywords:** manual chest compression, automated chest compression, out-of-hospital cardiac arrest, in-hospital cardiopulmonary resuscitation, night-time

## Abstract

Objective: We assessed the effectiveness of automated chest compression devices depending on the time of admission based on the frequency of iatrogenic chest injuries, the duration of in-hospital resuscitation efforts, and clinical outcomes among out-of-hospital cardiac arrest (OHCA) patients. Methods: We conducted a retrospective historical control study of OHCA patients in Japan between 2015–2022. The patients were divided according to time of admission, where day-time was considered 07:00–22:59 and night-time 23:00–06:59. These patients were then divided into two categories based on the in-hospital cardiopulmonary resuscitation (IHCPR) device: manual chest compression (mCC) group and automatic chest compression devices (ACCD) group. We used univariate and multivariate ordered logistic regression models adjusted for pre-hospital confounders to evaluate the impact of ACCD use during IHCPR on outcomes (IHCPR duration, CPR-related chest injuries, and clinical outcomes) in the day-time and night-time groups. Results: Among 1101 patients with OHCA (day-time, 809; night-time, 292), including 215 patients who underwent ACCD during IHCPR in day-time (26.6%) and 104 patients in night-time group (35.6%), the multivariate model showed a significant association of ACCD use with the outcomes of in-hospital resuscitation and higher rates of return in spontaneous circulation, lower incidence of CPR-related chest injuries, longer in-hospital resuscitation durations, greater survival to Emergency Department and hospital discharge, and greater survival with good neurological outcome to hospital discharge, though only in the night-time group. Conclusions: Patients who underwent ACCD during in-hospital resuscitation at night had a significantly longer duration of in-hospital resuscitation, a lower incidence of CPR-related chest injuries, and better outcomes.

## 1. Introduction

Out-of-hospital cardiac arrest (OHCA) remains a global public health problem throughout the world, including in Japan [1,2]. Although cardiopulmonary resuscitation (CPR) is essential to save patients with OHCA, it is well known that cardiopulmonary resuscitation (CPR) could be a cause of iatrogenic chest injuries (e.g., pneumothorax or rib fracture). Chest injuries could result in ineffective chest compressions because failure of thoracic compliance causes interruption of the cycle of positive and negative pressure [3,4]. It has also been reported that patients who experienced OHCA and received CPR during night-time had shorter resuscitation durations, higher incidence of CPR-related chest injuries, and lower survival rate [5] compared to day-time [6]. The higher incidence of CPR-related chest injuries may indirectly suggest that patients who received CPR during night-time received lower quality CPR, with inappropriate chest compressions, than those who received CPR during day-time.

High-quality chest compression is a pivotal part of resuscitation, and the effectiveness of manual chest compressions depends on the endurance of the resuscitation team [7]. To resolve these obstacles, mechanical chest compression devices were developed to maintain chest compression at appropriate rates and depths, and the devices reportedly produce no significantly different outcomes (compared to manual compressions) in terms of mortality and neurological outcomes after OHCA [8]. Although the Emergency Cardiovascular Care (ERC) and American Heart Association (AHA) resuscitation guidelines also consider mechanical chest compressions acceptable for CPR in unstable settings [9,10], no reports have examined the efficacy of automated chest compression devices according to treatment time period. Thus, this study aimed to evaluate the effectiveness of automated chest compression devices depending on the time of admission to the ED (daytime or nighttime) with respect to the outcomes of in-hospital cardiopulmonary resuscitation (IHCPR) duration, the frequency of iatrogenic chest injuries, the rates of return of spontaneous circulation (ROSC), and the survival and neurological outcomes among patients with OHCA.

## 2. Material and Methods

### 2.1. Study Design and Setting

This single-center retrospective historical control study was conducted at a tertiary emergency critical care center (Tokyo, Japan) that receives approximately 170 patients with OHCA each year. The number of day-time admissions was approximately 250% greater than the number of night-time admissions at the center. The medical records of OHCA patients transferred between 1 April 2015 and 31 March 2022 were surveyed. We provided the staffing shift during the study period as follows: (i) day-time (>2 board-certified emergency physicians, 3 medical interns, and 3 nurses); and (ii) night-time (1 board-certified emergency physician, 3 medical interns, and 2 nurses). All required procedures and medications were equally available during day-time and night-time and during weekdays and weekends. Pre-hospital factors were recorded as similar between day-time and night-time; the Japanese EMS system ensures a constant quality assessment of pre-hospital care (including holidays). At the study center, the method of chest compressions on arrival at the emergency department (ED) was formally changed on 1 May 2020, at which point automated mechanical compression replaced manual compression. During the study period, CPR and post-cardiac arrest care (including targeted temperature management) were provided consistently in accordance with the 2015 or 2020 AHA and ERC Guidelines for CPR. 

This study complied with the principles of the 1964 Declaration of Helsinki and its amendments. The study protocol was approved by the ethics committee of Tokyo Medical and Dental University Hospital of Medicine (M2022-249). All patient data were retrospectively collected from electronic medical charts and anonymized before statistical analyses.

### 2.2. Study Population

This study included consecutive OHCA patients who were transferred to the Tokyo Medical and Dental University Hospital between 1 April 2015, and 31 March 2022. We excluded patients who were less than 18 years old, had do-not-attempt-resuscitation orders, had cardiac arrest due to trauma, received open-chest CPR, did not undergo chest computed tomography (CT) examination, or had missing or insufficient data regarding the study variables (i.e., lack of information regarding CPR duration, witnessed status, or initial rhythm). In addition, patients who were not suitable for mechanical CPR (e.g., those with severe cachexia, morbid obesity, and chest wall deformity) were also excluded. Patients were excluded if they had received CPR using an automated chest compression device after ED arrival before 1 May 2020, regardless of setting, as this study aimed to evaluate if the quality of CPR performed by emergency medical staff is comparable to mechanical chest compression devices.

The Lund University Cardiac Assist System 3 (LUCAS 3, Stryker, Kalamazoo, MI, USA) was used as the mechanical chest compression device in our hospital for in-hospital resuscitation after 1 May 2020. The LUCAS 3 is a piston-based device that provides active compression and decompression via a suction cup placed at the center of the chest.

### 2.3. Data Collection

The following data were collected retrospectively from medical records: age, sex, ED admission time, presence or absence of a witness to the cardiac arrest, presence or absence of bystander CPR before paramedics’ arrival at the scene, shockable rhythm status, the cause of the cardiac arrest, whether or not ROSC was achieved, in-hospital CPR (IHCPR) duration, out-of-hospital CPR (OHCPR) duration, and status at hospital discharge (i.e., survival or death). We also collected data regarding CPR-related chest injuries (i.e., rib fractures, sternal fractures, pleural effusion/hemothorax, or pneumothorax) from the patients’ medical records and 64-slice computed tomography (CT) imagery. Our hospital has a general policy of routinely performing CT after CPR to identify the cause of cardiac arrest in non-traumatic cases. The CT findings were interpreted by ≥2 board-certified emergency physicians and one radiologist.

### 2.4. Definitions 

OHCA was defined as cardiac arrest that occurred out-of-hospital, with the patient being unresponsive to stimulation, gasping or not breathing, and having carotid arteries without a palpable pulse for a maximum assessment interval of 10 s, as previously described [11]. Based on the hospitals’ shift schedules and the results of previous studies [12], we divided the patients according to whether they were admitted to the emergency department (ED) during day-time (07:00–22:59) or night-time (23:00–06:59). OHCPR duration was defined as the interval between EMS dispatch and ED arrival in cases with bystander CPR [13], or as the interval between EMS arrival at the scene and ED arrival in cases without bystander CPR. IHCPR duration was defined as the interval between ED arrival and termination of resuscitation or ROSC [14]. For this study, ROSC is defined as the return of spontaneous circulation that lasted at least 5 min. We defined CPR-related chest injuries as rib fractures, sternal fractures, pleural effusion/hemothorax, or pneumothorax, as previously described [6].

### 2.5. Outcome Variables

The primary outcomes of this study were defined as the rate of ROSC, frequency of CPR-related chest injuries, and duration of IHCPR. We defined secondary outcomes as the rate of survival to ED discharge, the rate of survival to hospital discharge, and the rate of survival with good neurological outcomes to hospital discharge. Cerebral Performance Category (CPC) scores were used to classify neurological outcomes into five classes: (1) full recovery or mild disability; (2) moderate disability but independent in activities of daily living; (3) severe disability with dependence for support in activities of daily living; (4) persistent vegetative state; and (5) death. CPC scores of 1 or 2 indicate good neurological outcomes, while CPC scores of 3 or 4 indicate poor outcomes [15]. 

### 2.6. Statistical Analysis

Categorical variables are presented as numbers (percentages), while continuous variables are presented as medians (interquartile ranges), as appropriate. For univariate analysis, we used the Student t-test or Mann–Whitney U test to compare continuous variables and the χ^2^ test or Fisher’s exact test to compare categorical variables, as appropriate. 

We divided the enrolled patients according to whether they were admitted to the ED during daytime or nighttime, and further divided them into two categories based on the IHCPR technique used: the manual chest compression (mCC) group (from 1 April 2015 to 31 April 2020); and the automatic chest compression device (ACCD) group (from 1 May 2020 to 31 March 2022) in each group. First, we used a multivariate logistic regression model to evaluate the interaction between in-hospital ACCD use and admission time for the outcomes to evaluate whether the effect of in-hospital ACCD use was affected by admission time. We incorporated age, sex, witnessed status, bystander CPR status, initial rhythm (shockable or not), cause of cardiac arrest, and OHCPR duration as explanatory variables in the multivariate model, which were clinically plausible and well-known confounders in previous epidemiologic studies. Second, we used univariate and multivariate logistic regression models to evaluate the impact of ACCD use during IHCPR on outcomes in the day-time and night-time groups, respectively. Finally, factors that showed significant differences between day-time and night-time were incorporated into the multivariate model, and we assessed the multivariate analysis of variance between groups. All statistical analyses were performed using the R software (version 4.1.1; R Foundation for Statistical Computing, Vienna, Austria). Moreover, we used a command to add the statistical functions that are frequently used in biostatistics. All reported *p* values were two-sided, and *p* values < 0.05 were considered statistically significant.

## 3. Results

The patient selection process identified 1101 patients with OHCA, including 809 day-time cases (73.5%) and 292 night-time cases (26.5%). In our hospital, before 1 May 2020, all chest compressions administered to patients on arrival at the emergency department (ED) were performed by hand, and no patients received CPR on ED arrival using an automated chest compression device during the period. The day-time OHCA cases involved 215 patients (26.6%) who underwent automated chest compression devices during IHCPR, while the night-time cases involved 104 patients (35.6%) (Figure 1). The baseline characteristics of the day-time and night-time groups are summarized in Table 1. The age and proportion of females were similar between the day-time and night-time groups. The night-time group had relatively lower rates of witnessed OHCA, bystander CPR, and shockable initial rhythm, as well as a relatively longer duration of OHCPR. Table 2 provides the baseline characteristics of each patient group divided according to the admission time and IHCPR device. In both day-time and night-time groups, there were no significant differences between the mCC and ACCD groups in the rates of witnessed OHCA, bystander CPR, shockable initial rhythm, or duration of OHCPR. 

The results of the univariate analyses are summarized in Table 3. In the day-time group, there were no significant differences in the IHCPR duration and the rate of chest injuries between the mCC and ACCD groups. However, in the night-time group, the ACCD group had a significantly longer IHCPR duration (median 27 vs. 34 min) and a lower incidence of chest injuries (59.6 vs. 39.4%), relative to the mCC group. There were no significant differences in the rates of ROSC, survival to ED discharge, survival to hospital discharge, and survival with good neurological outcomes to hospital discharge between the mCC and ACCD groups in the day-time group. However, in the night-time group, the ACCD group had a significantly higher rate of ROSC (26.6% vs. 33.7%), survival to ED discharge (16.0% vs. 23.1%), survival to hospital discharge (6.9% vs. 12.5%), and survival with good neurological outcome at hospital discharge (4.3% vs. 8.7%).

In the entire study cohort, all *p* values for the interaction term of the admission time categories (day-time or night-time group) and in-hospital ACCD use were <0.001, which indicated a significant intra-group difference in the impact of in-hospital ACCD use on clinical outcomes (Appendix A). Table 4 presents the multivariate analysis results regarding the impact of in-hospital ACCD use on outcomes in the day-time and night-time groups. After adjusting for confounding factors, we found a significant association between night-time in-hospital ACCD use and longer IHCPR duration, lower chest injury rate, higher rate of ROSC, survival to ED discharge, survival to hospital discharge, and survival with good neurological outcome at hospital discharge. However, we did not observe a significant impact of in-hospital ACCD use on clinical outcome in the day-time group. All *p* values in the multivariate analysis of variance between groups were <0.001, indicating significant between-group differences in the outcomes.

### Limitations

First, this was a retrospective observational study conducted at a single hospital with a long study period and limited sample size; thus, there was a risk of residual confounding and type II error. Furthermore, we did not calculate the sample size, and this could affect the power of this study. Second, bystander characteristics, such as their occupation or whether they had medical training, were not identified. Third, detailed reasons (s) for termination of resuscitation in the ED were not always available. Fourth, we did not consider the impact of the COVID-19 pandemic on OHCA epidemiology. Although the COVID-19 pandemic did not appear to affect OHCA incidence or outcomes in our region—Tokyo, Japan [16] —OHCA studies from Europe and the United States reporting outcome events during the COVID-19 pandemic were consistent and documented lower rates of ROSC, less frequent survival to hospital admission, lower survival to hospital discharge, and worse neurological outcomes [17,18]. Furthermore, the rates of bystander CPR and neurologically favorable outcomes after OHCA reportedly decreased significantly during the COVID-19 pandemic in another region (Osaka, Japan) [19]. The timing and severity of COVID-19 outbreaks, behavioral restrictions such as lockdowns, and implementation of preventive measures varied across regions, even in Japan. Finally, although Utstein-style guidelines have recommended defining ROSC as return of spontaneous circulation for ≥30 s [20], this study defined ROSC as return of spontaneous circulation for ≥5 min, based on the absence of clinical data at 30-s intervals in the hospitals’ electronic medical record systems.

## 4. Discussion

In this retrospective control study, we evaluated the impact of in-hospital ACCD use, according to time of admission, on the clinical outcomes of 1101 patients with OHCA. The present study revealed that for night-time OHCA cases, the ACCD group had longer in-hospital resuscitation durations and a lower incidence of CPR-related chest injuries than the mCC group, even after adjusting for pre-hospital confounding factors. Notably, we found a higher impact of in-hospital ACCD use on survival and neurological outcomes in the night-time group than in the day-time group. These results suggest that the quality of CPR may be higher by using ACCD for in-hospital resuscitation than by manual chest compression at night. 

In contrast to several recent randomized clinical trials [21], a pilot study [22], systematic reviews, and meta-analyses [23,24], our results demonstrated improved survival rates with mechanical CPR compared to manual CPR controls in limited situations. The relatively small sample size in the present study might explain these differences. Another possible explanation could be related to the Japanese EMS system, in which there are restrictions on the decision to terminate resuscitation in the field [25,26]. For example, the in-hospital survival rate in the present study was 9.0%, which is much lower than previously reported rates [21,22]. 

The quality of manual chest compression could depend on several factors such as the practitioner’s skill, environment, and mental and physical strength [27,28]; this effectiveness could vary depending on the individual [29]. The fatigue and exhaustion of the practitioners due to prolonged CPR leads to a decrease in the effectiveness of CPR [13]. Therefore, it might be difficult to maintain high-quality CPR (i.e., appropriate hand position, depth, and rate) during night-time resuscitation. Mechanical compression devices have been invented to solve these problems generated from manual chest compression to standardize the CPR process and quality and to increase the effectiveness of chest compression [30]. ACCD is reported to produce no significantly different outcomes (vs. chest compressions performed by healthcare providers) in terms of survival rate and neurological outcomes after OHCA, to minimize pauses during transport, and to allow rescuers to focus on advanced life support [8]. Although the International Liaison Committee on Resuscitation’s (ILCOR) acknowledged the utility of mechanical devices in situations where high-quality manual chest compressions may be impractical or dangerous to rescuers [31], the routine use of automated chest compression devices has not yet been recommended [8]. Furthermore, the use of mechanical devices is also reportedly associated with an increased risk of chest injuries [32,33]; however, these findings should be interpreted with caution as they are prone to selection bias and the quality of manual chest compression delivered, as the comparator group, is generally not recorded. 

The delivery of high-quality manual chest compressions over a prolonged period is an aerosol-generating procedure that is physically exhausting. The guidelines of the American Heart Association and the European Resuscitation Council that recommend measures to be taken considering the COVID-19 pandemic have been published, [34,35] and they have recommended that, in settings with protocols in place and experts in their use, manual chest compressions should be replaced with mechanical CPR devices to reduce the number of rescuers. LUCAS3 was implemented at various stages of resuscitation and designed to minimize CPR interruptions. Therefore, a protocol requiring device implementation at a particular point of care may produce different results. For example, device application for patients who required a procedure, such as catheterization or chest tubes, appeared to be modestly beneficial [36]. Furthermore, in situations where high-quality manual chest compressions cannot be safely delivered or CPR performance is impeded by factors such as the COVID-19 pandemic or during night-time, using a mechanical device could be a reasonable clinical approach [37].

The findings of the present study are strengthened by the fact that all included patients underwent CT examinations, which allowed the detection of CPR-related chest injuries. The two study periods had similar settings regarding treatment guidelines, staffing, etc. Furthermore, we found for the first time that using a mechanical device for in-hospital resuscitation could also be a better clinical approach during night-time, in addition to the aforementioned unstable situations. 

In summary, this retrospective control study revealed that OHCA patients who underwent in-hospital resuscitation using an automated chest compression device at night had a significantly longer duration of in-hospital resuscitation, a significantly lower incidence of CPR-related chest injuries, and significantly better clinical and neurological outcomes. Further large-scale research is necessary to confirm the results of this study. 

## Figures and Tables

**Figure 1 jpm-13-01202-f001:**
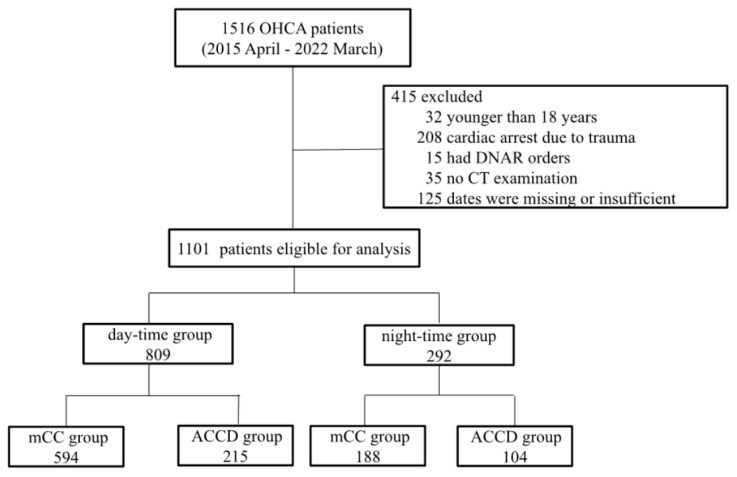
Patient flow diagram. OHCA, out-of-hospital cardiac arrest; DNAR, do-not-attempt-resuscitation; CT, computed tomography; mCC, manual chest compression; ACCD, automatic chest compression devices.

**Table 1 jpm-13-01202-t001:** Baselines characteristics of the day-time and night-time groups.

	Day-Time Group, N = 809	Night-Time Group, N = 292	*p* Value
Age, median [IQR]	68 [56–81]	66 [52–80]	0.483
Female, n (%)	239 (29.5)	90 (30.8)	0.393
Witnessed, n (%)	308 (38.1)	65 (22.3)	<0.001
Bystander CPR, n (%)	191 (23.6)	43 (14.7)	0.014
Shockable Initial rhythm, n (%)	129 (15.9)	36 (12.3)	0.003
Cardiogenic, n (%)	163 (20.1)	58 (19.9)	0.321
OHCPR duration, min, median [IQR]	29 [23–35]	37 [28–42]	<0.001

Abbreviations: IQR, interquartile range; OHCPR, out-of-hospital cardiopulmonary resuscitation; mCC, manual chest compression; ACCD, automatic chest compression devices.

**Table 2 jpm-13-01202-t002:** Baselines characteristics of each patient group according to the admission time and the device of IHCPR.

	Day-Time Group, N = 809			Night-Time Group, N = 292		
	mCC Group N = 594	ACCD Group N = 215	*p* Value	mCC Group N = 188	ACCD Group, N = 104	*p* Value
Age, median [IQR]	68 [55–81]	65 [57–83]	0.395	65 [48–79]	65 [59–79]	0.683
Female, n (%)	180 (30.3)	59 (27.4)	0.543	59 (31.4)	31 (29.8)	0.793
Witnessed, n (%)	236 (39.7)	72 (33.5)	0.106	44 (23.4)	21 (20.2)	0.261
Bystander CPR, n (%)	149 (25.1)	42 (19.5)	0.111	28 (14.9)	15 (14.4)	0.391
Shockable Initial rhythm, n (%)	90 (15.1)	39 (18.1)	0.277	24 (12.8)	12 (11.5)	0.754
Cardiogenic, n (%)	115 (19.4)	48 (22.3)	0.171	37 (19.7)	21 (20.2)	0.791
OHCPR duration, min, median [IQR]	28 [23–35]	29 [24–36]	0.261	34 [26–40]	35 [31–42]	0.231

Abbreviations: IQR, interquartile range; OHCPR, out-of-hospital cardiopulmonary resuscitation; mCC, manual chest compression; ACCD, automatic chest compression devices.

**Table 3 jpm-13-01202-t003:** Univariate analysis of primary and secondary outcomes.

Outcomes	Day-Time Group N = 809			Night-Time Group N = 292		
	mCC N = 594	ACCD N = 215	*p* Value	mCC N = 188	ACCD N = 104	*p* Value
ROSC, n (%)	191 (32.2)	78 (37.1)	0.031	49 (26.6)	35 (33.7)	<0.001
Chest injuries, n (%)	202 (34.0)	81 (37.7)	0.328	112 (59.6)	41 (39.4)	<0.001
Rib fractures, n (%)	145 (24.4)	64 (30.0)	-	88 (46.8)	25 (24.0)	-
Sternal fractures, n (%)	82 (13.8)	35 (16.3)	-	56 (30.0)	11 (10.6)	-
Pleural effusion/hemothorax, n (%)	56 (9.4)	25 (11.6)	-	45 (23.9)	10 (9.6)	-
Pneumothorax, n (%)	70 (11.8)	31 (14.4)	-	52 (27.7)	14 (13.5)	-
IHCPR duration, min [IQR]	35 [28–39]	34 [27–38]	0.762	27 [24–32]	34 [28–38]	<0.001
Survival to ED discharge, n (%)	108 (18.2)	42 (19.5)	0.271	30 (16.0)	24 (23.1)	<0.001
Survival to hospital discharge, n (%)	51 (8.6)	22 (10.2)	0.106	13(6.9)	13 (12.5)	<0.001
Survival with good neurological outcome to hospital discharge, n (%)	39 (6.6)	16 (7.4)	0.214	8 (4.3)	9 (8.7)	0.021

ROSC, return of spontaneous circulation; IHCPR, in-hospital cardiopulmonary resuscitation; IQR, interquartile range; ED, emergency department; mCC, manual chest compression; ACCD, automatic chest compression devices.

**Table 4 jpm-13-01202-t004:** Impact of ACCD use during IHCPR on outcomes in multivariate regression analysis.

	Day-Time Group			Night-Time Group		
	AOR [95% CI]	AD [95% CI]	*p* Value	AOR [95% CI]	AD [95% CI]	*p* Value
ROSC	0.66 [0.24–1.42]	-	0.354	1.14 [1.05–1.37]	-	<0.001
Chest injuries	0.78 [0.51–1.21]	-	0.431	0.41 [0.30–0.81]	-	<0.001
IHCPR duration	-	−1.2 [−2.1–0.9]	0.526	-	6.1 [4.5–7.5]	<0.001
Survival to ED discharge	0.81 [0.43–1.42]	-	0.651	1.13 [1.04–1.27]	-	<0.001
Survival to hospital discharge	0.89 [0.51–1.52]	-	0.472	1.10 [1.03–1.21]	-	<0.001
Survival with good neurological outcome to hospital discharge	0.94 [0.53–1.48]	-	0.235	1.09 [1.04–1.12]	-	0.002

Abbreviations: AOR, adjusted odds ratio; AD, adjusted difference; CI, confidence interval; IHCPR, in-hospital cardiopulmonary resuscitation; ROSC, return of spontaneous circulation.

## Data Availability

Raw data were generated at Tokyo Medical and Dental University Hospital of Medicine. Derived data supporting the findings of this study are available from the corresponding author (WT) on request.

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
