# Peer review of "Manual Chest Compression versus Automated Chest Compression Device during Day-Time and Night-Time Resuscitation Following Out-of-Hospital Cardiac Arrest: A Retrospective Historical Control Study"

_jpm, 2023, doi:10.3390/jpm13081202_

Round 1
Reviewer 1 Report
Dear authors,
The manuscript aimed to identify the effect of ACCD on OHCA according to the treatment time period by using a retrospective study. I think the study design was very appropriate to measure such results. However, I have some comments as follows;
Introduction: Please elaborate more on the reason why you choose to examine the differences of treatment time periods, any specific reason?
Method: Please describe more on the differences between day and night period other than staff on shifts.
Have you considered sample size calculation? in order to explain the power of study.
What were the reasons behind choosing these period (2015-2022)?
Please clarify the reason why patients who received ACCD before May 1,2020 were excluded. I think the reason given was not justified. (Line "In addition, patients were excluded if they had received CPR using an automated chest compression device after ED arrival before May 1, 2020, regardless of setting, as this study aimed to evaluate if the quality of CPR performed by emergency medical staff is comparable to mechanical chest compression devices."). Moreover, if these group were excluded, it would mean that all recruited patients in the study in ACCD group were before May 1, 2020 which would make the study less credible since there might be some staff's , procedures' , other factors' differences between before and after May 1,2020.
Results: in table one, there were two parameters that were significantly differences between day and nighttime period which are witness arrest and OHCPR duration, mean min [IQR]. Did you consider putting these parameters in the multivariate analysis as well?
Limitation: please give more information about the differences of OHCA and CPR management during COVID-19 pandemic to other period. Since AHA and ERC guidelines during COVID-19 pandemic have several different procedures, the information about Japanese guideline during the period may help the readers to understand the results more. If there were a lot of changes in practices during the COVID-19 period, the results that stated about differences in outcomes between day and nighttime will not be reliable. (ex. "Notably, we found a higher impact of in-hospital ACCD use on survival and neurological outcomes in the nighttime group than in the day-time group.")
Discussion: the paragraph that start with 'The quality of manual chest compression could depend on several factors such as the practitioner’s skill, environment, and mental and physical strength; this effectiveness could vary depending on the individual.', I am not sure what point the authors tried to make which information were from the results of this study and which were from literature. They looked like a summarization of literature which were also repetition with the introduction.
The manuscript was written in high quality of English academic language.
Author Response
# Reviewer 1
Introduction: Please elaborate more on the reason why you choose to examine the differences of treatment time periods, any specific reason?
Response:
Thank you for your comment. As per your suggestion, we have revised the Introduction section (page 2, lines 14-17).
Method: Please describe more on the differences between day and night period other than staff on shifts.
Response:
We appreciate your comment. As per your suggestion, we have mentioned the number of patients admitted during the night-time and day-time in the Material and Methods section (page 2, lines 43-45).
Have you considered sample size calculation? in order to explain the power of study.
Response:
Sample size calculation is usually performed in prospective randomized controlled studies. In retrospective studies, one would use statistical power instead of sample size calculation. Therefore, we have not calculated the sample size in this study. However, as you noted, this can affect the power of the study. This is an important confounding factor, and we would like to revisit this topic in a future large-scale study or randomized controlled trial. We have mentioned this limitation in the manuscript (page 9). Thank you for bringing it to our attention.
What were the reasons behind choosing these period (2015-2022)?
Response:
Thank you for your query. The care of patients with out-of-hospital cardiac arrest has improved over the past decade, and the relevant guidelines have been revised after 2015. To avoid the influence of the time period and include as many patients as possible, we set the study period to 2015–2022 (8 years). We have added this information to the Limitationsection (page 9).
Please clarify the reason why patients who received ACCD before May 1,2020 were excluded. I think the reason given was not justified. (Line "In addition, patients were excluded if they had received CPR using an automated chest compression device after ED arrival before May 1, 2020, regardless of setting, as this study aimed to evaluate if the quality of CPR performed by emergency medical staff is comparable to mechanical chest compression devices."). Moreover, if these group were excluded, it would mean that all recruited patients in the study in ACCD group were before May 1, 2020 which would make the study less credible since there might be some staff's , procedures' , other factors' differences between before and after May 1,2020.
Response:
Thank you for your valuable comment. In our hospital, before May 1, 2020, all chest compressions administered to patients on arrival at the emergency department (ED) were performed by hand, and no patients received CPR on ED arrival using an automated chest compression device during the period. As you noted, our description was confusing. Therefore, we have revised the Results section (page 5).
Results: in table one, there were two parameters that were significantly differences between day and nighttime period which are witness arrest and OHCPR duration, mean min [IQR]. Did you consider putting these parameters in the multivariate analysis as well?
Response:
Thank you for your question. As you noted, witnessed status and OHCPR duration showed significant differences between day-time and night-time. Therefore, we incorporated them into the multivariate model. We have revised the Statistical analysis section to better reflect this (page 5).
Limitation: please give more information about the differences of OHCA and CPR management during COVID-19 pandemic to other period. Since AHA and ERC guidelines during COVID-19 pandemic have several different procedures, the information about Japanese guideline during the period may help the readers to understand the results more. If there were a lot of changes in practices during the COVID-19 period, the results that stated about differences in outcomes between day and nighttime will not be reliable. (ex. "Notably, we found a higher impact of in-hospital ACCD use on survival and neurological outcomes in the nighttime group than in the day-time group.")
Response:
As per your suggestion, we have added a reference (reference 19 and revised the Limitation section (page 10, lines 7-13).
Discussion: the paragraph that start with 'The quality of manual chest compression could depend on several factors such as the practitioner’s skill, environment, and mental and physical strength; this effectiveness could vary depending on the individual.', I am not sure what point the authors tried to make which information were from the results of this study and which were from literature. They looked like a summarization of literature which were also repetition with the introduction.
Response:
We agree with you that the statement you quote in your comment was vague and difficult to understand. To rectify this, we have added a reference (Reference number 27, 28) and revised the Discussion section (page 10).

Reviewer 2 Report
This is a well designed study and a well written report on a topic of interest to the resuscitation community.
Issues with the report are indicated in the attachment. None are major.
The text confuses non-inferiority with lack of finding a statistical difference in several places.
Some citations do not support the statements they are attached to, possibly in part due to misnumbering the citations - the authors are encouraged to carefully review the citations. They should also avoid reliance on guidelines texts unless it is the guidelines themselves they are citing. For studies cited in the guidelines as support, the authors should cite the original studies.
It would be interesting to provide more granular detail about forms of damage. Means ± standard deviations are described, but averages for continuous variables are accompanied only by interquartile ranges (IQRs), suggesting that only medians have been used. This should be clarified. Interquartile range should be spelled out upon introduction of the acronym IQR.
The multivariate analysis is described as "ordered" logistic regression, which implies the use of ordinal variables, which is not evident. Either "ordered" should be omitted or the ordinal variables explained.
A correction for multiple comparisons in the multivariate analysis may be appropriate.
The large difference in rate of usage of ACCD in the day and in the night shifts calls for further examination and discussion.
Some additional minor points and suggestions for expression are in the attached markup.

The English language expression is generally quite clear. A few suggestions are in the marked up attachment.
Author Response
#Reviewer 2
The text confuses non-inferiority with lack of finding a statistical difference in several places.
Response:
Thank you for pointing this out. We have made the necessary corrections throughout the manuscript.
Some citations do not support the statements they are attached to, possibly in part due to misnumbering the citations - the authors are encouraged to carefully review the citations. They should also avoid reliance on guidelines texts unless it is the guidelines themselves they are citing. For studies cited in the guidelines as support, the authors should cite the original studies.
Response:
Thank you for your comment. We apologize for the error. We have reviewed and revised the references.
It would be interesting to provide more granular detail about forms of damage.
Response:
As per your recommendation, we have added information on the types of chest injuries to Table 3.
Means ± standard deviations are described, but averages for continuous variables are accompanied only by interquartile ranges (IQRs), suggesting that only medians have been used. This should be clarified. Interquartile range should be spelled out upon introduction of the acronym IQR.
Response:
Thank you for your valuable comment. We apologize for the error. The word “mean” was corrected to “median,” and we have revised Tables 1 and 2.
The multivariate analysis is described as "ordered" logistic regression, which implies the use of ordinal variables, which is not evident. Either "ordered" should be omitted or the ordinal variables explained.
Response:
Thank you for your comment. We have omitted the term “ordered.”
A correction for multiple comparisons in the multivariate analysis may be appropriate.
Response:
Thank you for your suggestion. Accordingly, we have assessed the multivariate analysis of variance (MANOVA) between groups. We have revised the Statistical analysis (page 5) and Results (page 9) sections to reflect this.
The large difference in rate of usage of ACCD in the day and in the night shifts calls for further examination and discussion.
Response:
At our institution, the method of performing chest compressions on arrival at the emergency department (ED) was formally changed on May 1, 2020; from this point onward, automated mechanical compressions were performed instead of manual compressions. Therefore, all patients brought to the ED before this date received manual chest compression, whereas those brought to the ED after this date received automated mechanical compressions during in-hospital resuscitation. However, as you noted, there were large difference between the rates of day-time and night-time ACCD use. This factor could affect the power of our study. Because this was a retrospective cohort study with a limited sample size, the groups could not be made in a way that the patients’ background was similar between the groups. We have emphasized this fact in the Limitation section (page 9).
Some additional minor points and suggestions for expression are in the attached markup.
Response:
Thank you very much for carefully checking our manuscript. We have revised the manuscript according to your suggestions.
